# *Gigaspora roseae* and *Coriolopsis rigida* Fungi Improve Performance of *Quillaja saponaria* Plants Grown in Sandy Substrate with Added Sewage Sludge

**DOI:** 10.3390/jof11010002

**Published:** 2024-12-24

**Authors:** Guillermo Pereira, Diyanira Castillo-Novales, Cristian Salazar, Cristian Atala, Cesar Arriagada-Escamilla

**Affiliations:** 1Departamento de Ciencias y Tecnologia Vegetal, Campus Los Ángeles, Universidad de Concepción, Juan Antonio Coloma 0201, Casilla 341, Los Ángeles 4451032, Chile; gpereira@udec.cl (G.P.); crisalazar@udec.cl (C.S.); 2Laboratorio de Microbiología Molecular y Biotecnología Ambiental, Departamento de Química, Centro de Biotecnología, Universidad Técnica Federico Santa María, Avenida España 1680, Valparaíso 2390123, Chile; diyaniracastillonovales@gmail.com; 3Escuela de Agronomía, Facultad de Ciencias Agronómicas y de los Alimentos, Pontificia Universidad Católica de Valparaíso, San Francisco s/n La Palma, Quillota 2260000, Chile; 4Instituto de Biología, Facultad de Ciencias, Pontificia Universidad Católica de Valparaíso, Campus Curauma, Avenida Universidad 330, Valparaíso 8331150, Chile; cristian.atala@pucv.cl; 5Laboratorio de Biorremediación, Departamento de Ciencias Forestales, Facultad de Ciencias Agropecuarias y Medioambiente, Universidad de la Frontera, Casilla 54-D, Temuco 4811230, Chile

**Keywords:** plant nursery, mycorrhizal and saprophytic fungi, residual liquid sludge, plant development

## Abstract

The use of living organisms to treat human by-products, such as residual sludge, has gained interest in the last years. Fungi have been used for bioremediation and improving plant performance in contaminated soils. We investigated the impact of the mycorrhizal fungus (MF) *Gigaspora roseae* and the saprophytic fungus (SF) *Coriolopsis rigida* on the survival and growth of *Quillaja saponaria* seedlings cultivated in a sandy substrate supplemented with residual sludge. *Q. saponaria* is a sclerophyllous tree endemic to Chile, known for its high content of saponins. We inoculated plants with the MF, the SF, and a combination of both (MF + SF). Following inoculation, varying doses of liquid residual sludge equivalent to 0, 75, and 100% of the substrate’s field capacity were applied. After 11 months, we found a positive influence of the utilized microorganisms on the growth of *Q. saponaria*. Particularly, inoculation with the SF resulted in higher plant growth, mycorrhizal colonization percentage, and higher enzymatic activity, especially after the application of the sludge. This increase was more evident with higher doses of the applied sludge. These results highlight the potential of combined microorganism and residual sludge application as a sustainable strategy for enhancing plant growth and reducing waste.

## 1. Introduction

Productive activity stands as a cornerstone of a country’s economic development. Nonetheless, the generation of waste and the excessive exploitation of natural resources can evolve into persistent agents of environmental deterioration [1,2]. Every year we generate close to 2.01 billion metric tons of municipal solid waste worldwide [3]. This is especially worrying considering global population growth and the poor management of these residues, especially in developing countries [4]. The disposal of urban organic waste currently ranks among the most pressing environmental concerns [5,6]. Thus, addressing the challenge of managing biosolids generated in urban centers represents a paramount ecological imperative worldwide. The agricultural or forestry valorization of biosolids from wastewater treatment plants emerges as a pragmatic solution, providing vital nutrients to crops and rationalizing the use of these waste resources [7,8,9]. Duggan and Wiles [10] and Rasiah et al. [11] supported the notion that the application of biosolids and municipal solid waste enriches soil properties, enhancing its physical and chemical attributes, thus promoting improved soil structure, porosity, permeability, and water retention, ultimately benefiting vegetation through increased soil nutrient reserves [9,12].

Nevertheless, this nutrient-rich organic matter derived from sludge presents a suite of challenges, including the impediment of direct application to agricultural soil due to its heavy metal content, presence of pathogens, and toxic organic compounds [13]. Additionally, the use of sewage sludge as a soil amendment raises concerns about potential environmental and human health risks associated with the presence of persistent organic pollutants (POPs), including polycyclic aromatic hydrocarbons (PAHs) and polychlorinated biphenyls (PCBs) [14]. Studies, however, have suggested that the utilization of microorganisms, such as mycorrhizal fungi, may serve as effective mechanisms for heavy metal sequestration, thereby reducing the translocation of elements to a plant’s aerial parts [15,16,17]. Moreover, these microorganisms may act as biosorbents for heavy metals, offering a potential alternative to conventional methods for the detoxification and recovery of toxic metals present in industrial wastewater [18]. These findings highlight the potential of mycorrhizal fungi in addressing heavy metal pollution in soil and water, offering promising avenues for sustainable environmental management. Furthermore, recent advancements in bioremediation techniques, such as phytoremediation and microbial-assisted remediation, have shown promise in effectively mitigating the adverse impacts of heavy metal contamination in various environmental matrices [19]. These approaches capitalize on the synergistic interactions between plants, microorganisms, and soil components to enhance the removal and immobilization of heavy metals, thus promoting ecosystem restoration and environmental sustainability. Not only mycorrhizal fungi have been used in bioremediation and treatment of polluted soils. Some saprophytic fungi can also contribute to heavy metal accumulation and immobilization [20,21] and bioremediation of other contaminants [22,23]. Thus, both mycorrhizal and saprophytic fungi could serve as a biotechnological tools for bioremediation of soils. Moreover, these types of fungi could act synergistically to improve plant growth in contaminated soil by reducing oxidative stress [24].

In this context, the treatment of urban waste and its subsequent utilization as biofertilizers in agriculture and forestry, in conjunction with appropriate soil microorganism management, holds promise as an effective and environmentally friendly approach enabling agroforesters to achieve satisfactory productivity outcomes in the field and contribute to urban waste managements and reduce pollution. Hence, the objective of this study is to evaluate the influence of the mycorrhizal fungus *Gigaspora roseae* and the saprophytic fungus *Coriolopsis rigida* on the survival and growth of *Quillaja saponaria* seedlings cultivated in a sandy substrate with the addition of sewage sludge. We used *Q. saponaria* as a model plant since (i) it is an endemic tree that grows in Central Chile, (ii) it is easy to obtain seedlings since it is currently produced in nurseries in large numbers for ornamental, medicinal, and forestry purposes, and (iii) it is relatively easy to grow and can grow rapidly compared to other native trees.

The saponin present in *Q. saponaria* is ecologically and industrially relevant due to its bioactive and pharmacological properties, such as hypolipidemic, anti-inflammatory, expectorant, and androgenic activities, making it widely used in the food and pharmaceutical industries. Saponins act as natural detergents, and have been studied for their physicochemical properties (e.g., as surfactants) and biological properties (e.g., as biocides and antibacterial agents), serving as key components in applications such as encapsulation agents, stabilizers for bioactive compounds, and foaming agents in beverages and processed foods [25,26,27]. In the agricultural industry, saponins play an important role as biocontrol agents due to their insecticidal, antimicrobial, and plant growth-promoting properties. Their exogenous application has been observed to accelerate seed germination, stimulate chlorophyll biosynthesis, and regulate cellulose synthesis in various plant species. Additionally, saponins extracted from *Q. saponaria* are used in veterinary vaccine formulations and as immunological adjuvants [28,29,30,31]. In the environmental sector, saponins are effective in soil remediation and the removal of heavy metals, such as cadmium, copper, and lead, due to their surfactant properties. These applications highlight their role not only as useful compounds in industrial processes but also as sustainable solutions for agriculture and environmental management [30,32]. For these reasons, the use of *Q. saponaria* in this study is particularly relevant, as its saponin content not only supports advancements in agricultural sustainability but also holds significant potential for innovations in biotechnology and environmental management [33,34]. Additionally, overexploitation of *Q. saponaria* bark has caused severe ecological damage to the species, and the bark has become a relatively scarce resource [35,36]. This sustainability issue has led to establishment of plantations where all the tree biomass is used, considering that leaves are close to 27.8% of the total weight, with a close to 3% saponin content [36]. Since productivity and total tree biomass is relevant to saponin yield, the use of benefic microorganisms and other methods, such as the addition of nutrient-enriched sludge which enhances growth and survival, could be essential for *Q. saponaria* cultivation in degraded and low fertility soils. Moreover, mycorrhizal and saprophytic fungi could interact with added sludge by increasing bioavailable nutrients for the plant and their own growth.

## 2. Materials and Methods

### 2.1. Microorganisms and Plants

We used the mycorrhizal fungus (MF) *Gigaspora roseae* (T.H. Nicolson & N.C. Schenck) and the saprophytic fungus (SF) *Coriolopsis rigida* (Berk. Et Mont.) Murrill, both sourced from the fungal collection maintained by the Bioremediation Laboratory at the Universidad de la Frontera, Temuco, Chile. The plant material selected for experimentation consisted of *Quillaja saponaria* Mol. (commonly known as Quillay). Seeds were harvested from mature trees in the city of Los Ángeles, Biobío Province, Chile, and subsequently stored at 4 °C.

### 2.2. Sludge and Substrates

The urban sludge utilized in this study was sourced from the secondary aerobic treatment of wastewater at the Vilcún plant, Aguas Araucanía S.A, located in the IX Region (Table 1). Sludge application to plants was conducted using aqueous extracts derived from the separation of liquid and solid fractions of residual sludge via orbital extraction [13]. For seedling germination, vermiculite served as the substrate. Following germination, soil from the Arenales Series was employed, sampled from the superficial horizon (20 cm depth) in the María Dolores aerodrome sector of the Biobío Province, VIII Region (37°22’41″ S–72°25’30″ W). This substrate underwent sterilization in an autoclave (121 °C for one hour) for three consecutive days.

### 2.3. Biochemical Determinations

The rhizosphere soil samples were collected after harvest by shaking the roots gently. β-glucosidase activity was determined by measuring *p*-nitrophenol (PNP) released from *p*-nitrophenyl-β-D-glucopyranoside (PNG) according to the method of Eivazi and Tabatabai [37]. The amount of *p*-nitrophenol was determined at 398 nm and expressed as µmol PNP/g dry soil/h. The fluorescein diacetate (FDA) hydrolysis was performed as described by previously [38]. Laccase activity was assayed with 5 mM DMP (2,6-dimethoxyphenol) in 100 mM of acetate buffer pH 5.0 (ε469 = 27,500 M^−1^ cm^−1^) [39]. Mn-dependent peroxidase (MnP) was estimated by the formation of Mn+3–tartrate complex as described by Saparrat and Guillén [39]. International enzymatic units (μmol of oxidized product per minute) were used.

### 2.4. Plant Propagation

Prior to sowing, Quillay seeds were soaked in drinking water for 24 h before being planted in two 1500 cm^3^ pots filled with vermiculite. These containers were then placed in a growth chamber set to a temperature of 24 ± 1 °C, with the substrate moisture maintained close to field capacity. Once the Quillay seeds germinated and developed their second pair of true leaves, seedlings were transplanted into 250 cc containers and inoculated with the corresponding fungi using previously sterilized sandy soil as the growth substrate.

### 2.5. Cultivation of Mycorrhizal and Saprophytic Fungi

For the cultivation and propagation of the mycorrhizal fungus *G. roseae*, *Trifolium pratense* L. (Clover) plants were utilized. The “Trap Plant” method was employed, ensuring abundant inoculum both in the rhizosphere soil and on the roots [40,41]. The mycorrhizal inoculum consisted of a mixture of 2 g rhizosphere soil of *Medicago sativa* L. root fragments and spores per pot. These amounts were predetermined to achieve high levels of root colonization [40,41]. To prepare the *C. rigida* inoculum, wheat (*Triticum aestivum* L.) seeds were inoculated with a 1 cm^2^ disk of MEA withdrawn from 14-day-old fungal cultures grown at 28 °C. Ten wheat seeds colonized by the mycelium of the fungi were added to the soil in each pot. These preparation and soil inoculation conditions were previously reported [42,43,44].

### 2.6. Plant Inoculation Experiment

Inoculation involved applying two grams of MF and SF inoculum separately, or a mixture of both (one gram of each fungus simultaneously) in the lower third of the containers during transplanting. Liquid sludge application occurred ten weeks after transplanting and inoculation to ensure effective plant colonization. We applied three sludge doses: 0%, 75%, and 100% of substrate field capacity, corresponding to 0, 22.5, and 30 mL doses. Plant growth in height was measured twice, at sludge application and trial end, while diameter at collar (DAC) was determined at trial end. Aerial and root biomass were obtained by cutting plants at the DAC level, extracting roots, and drying samples at 60 °C for 2 days before measurements of dry weight using an analytical scale. A root staining methodology [45] was used to assess plant colonization, and the degree of mycorrhization in *Q. saponaria* was evaluated using Giovannetti and Mosse’s [46] method.

### 2.7. Statistical Analysis

The study was designed as a factorial analysis incorporating two factors: microorganisms and sludge dose, with four levels of microorganism inoculation and three levels of sludge dose. Each treatment consisted of 6 replicates. Statistical analysis was conducted using a two-way analysis of variance (2-way ANOVA), and significant differences between treatments were identified using the Tukey test for multiple comparisons [47]. Statistical analysis was performed using Statistica software version 6.0.

## 3. Results

### 3.1. Impact of G. roseae and C. rigida on the Growth of Q. saponaria Seedlings

The study’s findings reveal a positive influence of the utilized microorganisms (MF and SF) on the height and diameter growth of *Q. saponaria* plants, observed eleven months post-application. Figure 1A illustrates that the most favorable response in height was observed in plants treated with *C. rigida* or in combination with *G. roseae*. Statistically significant differences were evident between these treatments and plants inoculated solely with the MF *G. roseae*, as well as compared to control plants, which exhibited the least height growth. Inoculation with *C. rigida* increased plant height by 337.5% and the MF + SF treatment by 167.5% compared to control plants. The results regarding the diameter variable (Figure 1B) exhibit a pattern like that observed for height. Plants treated with *C. rigida* attained the largest diameter, followed by those treated with the MF + SF combination. In contrast, control plants exhibited the lowest growth in diameter. These findings indicate significant differences between the treatment with *C. rigida* and both the MF *G. roseae* treatment and the control, as well as significant statistical differences between the MF + SF treatment and the MF *G. roseae* treatment compared to the control.

### 3.2. Impact of Tested Fungi and Residual Sludge % on Shoot and Root Biomass in Q. saponaria Seedlings

We found a positive effect of MF and SF on both the aerial and root biomass of *Q. saponaria* plants after 11 months (Figure 2, two-way ANOVA, inoculation factor *p* < 0.05 for both aerial and root biomass). This effect was evident not only when these fungi were applied individually, but also when they were co-inoculated. Notably, the treatment with the SF *C. rigida* resulted in the most significant aerial growth in both evaluated variables (Figure 2, Tukey test *p* < 0.05), followed by the MF + SF combination and the MF *G. roseae* treatment. Conversely, the control treatment exhibited the lowest aerial and root biomass growth (Figure 2, Tukey test, *p* < 0.05). Furthermore, a clear correlation between aerial and root growth of *Q. saponaria* plants was evident (Figure 2). Similar to the shoot biomass, the treatment with *C. rigida* demonstrated the highest root growth, while the control treatment showed the lowest (Figure 2, Tukey test *p* < 0.05). In contrast, the *G. roseae* and MF + SF treatments did not statistically differ from control plants.

An increase in sludge dose (0, 75, and 100%) affected the aerial biomass of *Q. saponaria* (Figure 2, two-way ANOVA, dose factor *p* < 0.05) but each sludge dose had a different, non-linear, effect (*p* < 0.05 in the inoculation x dose interaction of the two-way ANOVA). Sludge dose, however, only marginally affected root biomass (Figure 2, two-way ANOVA, *p* = 0.0577 for dose factor). Treatment with *C. rigida* showed significantly higher aerial biomass (as stated above), with the 100% sludge dose yielding the highest observed values throughout the trial. This difference was statistically significant when compared to the *C. rigida* treatments with 75% and 0% sludge doses, as well as with the other treatments assessed (Tukey test *p* < 0.05). Conversely, Figure 2 illustrates that the biomass of *Q. saponaria* was favored by treatment with the MF *G. roseae* and the MF + SH combination up to the 75% sludge dose, with growth decreasing at higher doses (100% sludge). This decrease was particularly pronounced when the fungi combination was used, affecting both aerial and root growth. Notably, the control treatment (without microorganism application) exhibited the lowest below- and above-ground biomass (Figure 2).

### 3.3. Analysis of Mycorrhizal Colonization Achieved in Q. saponaria Plants

Inoculation affected mycorrhizal colonization percentage in *Q. saponaria* (two-way ANOVA inoculation factor, *p* < 0.05), but sludge dose did not (two-way ANOVA dose factor, *p* > 0.05). The inoculation x dose interaction was statistically significant (two-way ANOVA, *p* < 0.05), indicating that dose did not affect each inoculation treatment equally. Overall, plants inoculated only with the MF *G. roseae* evidenced the highest degree of mycorrhization, particularly without residual sludge application (Figure 3), although it did not statistically differ from the other sludge doses. In contrast, when *Q. saponaria* plants were treated with the MF + SF combination, their degree of mycorrhization appeared to benefit from adding residual sludge (Figure 3). These plants achieved the highest degrees of mycorrhization with 75 and 100% sludge doses compared to 0% (Tukey test, *p* < 0.05).

### 3.4. Chemical Characteristics of Soil Substrate Before and After Residual Sludge Application

Table 1 presents the chemical characteristics of the sewage sludge, and the sandy soil before and after the application of increasing doses of residual sludge (0, 75 y 100%). The Table 2 show that the organic sludge contained the highest percentage of organic material (52.1%), while the sandy soil has negligible organic content (0.03%). Soil mixes with sludge showed a slight increase in organic matter content compared to sandy soils. The original organic sludge contained the highest Kjeldahl nitrogen content (38.7 g kg^−1^), while the sandy soil had significantly lower nitrogen content (6.0 g kg^−1^). Soil mixed with sludge exhibited a progressive increase in nitrogen and Olsen-P content as sludge proportions increased. Available K (mg kg^−1^), cation exchange capacity (CEC cmol kg^−1^), Fe (mg kg^−1^), Mn (mg kg^−1^), Zn (mg kg^−1^), and Cu (mg kg^−1^): Levels were higher in the original organic sludge and decreased in sandy soils and soil mixes with sludge. However, slight variations in these micronutrient levels existed among different soil mixes. Exchangeable Al, exchangeable Mg (mg kg^−1^): Exchangeable aluminum levels are relatively low in all substrates, with minimal variation observed.

### 3.5. Enzymatic Activity

We observed an increase in laccase and MnP activity in vitro with increasing sludge dose from 0 to 75% in plants inoculated with the SF (Table 3). However, at 100% of the sludge dose, there was no enzymatic activity detected (Table 3). The hydrolysis of FDA was promoted when inoculating with the tested fungi, both in single inoculations and in co-inoculation. However, adding sewage sludge negatively affected FDA activity (Table 4). Contrarily, the β-glucosidase activity showed that the inoculation with *G. roseae* increased in all treatments tested, except when co-inoculated with *C. rigida*. In this case, a significant increase was observed only at 75% sludge dose (Table 5).

## 4. Discussion

There is growing recognition of the potential environmental and socio-economic benefits of applying a circular approach to urban organic waste management through resource recovery [48,49]. For example, the application of residual sludge in forest crops and reforestation tasks has garnered attention as a potential solution for enhancing soil fertility and promoting plant growth, particularly in nutrient-deficient environments [9,50]. Recent studies have highlighted the positive effects of sludge application on soil structure, nutrient availability, and plant growth, emphasizing its role in sustainable agricultural practices [51]. Moreover, Smith et al. [52] demonstrated the effectiveness of sludge-based fertilizers in improving crop yield and reducing the environmental impact of conventional chemical fertilizers. We found that the addition of different sludge doses (0, 75, and 100%) improved the performance of *Q. saponaria* seedlings after 11 months. The inoculation with the SF *C. rigida* stands out for its remarkable growth benefits, particularly evident with the 100% sludge dose, which yielded the highest aerial and root final biomass observed throughout the trial. We found that inoculation with *G. roseae* and the combination of both fungi yielded favorable growth outcomes up to the 75% sludge dose. However, growth showed a decrease at higher doses, which was particularly pronounced when employing the MF + SF combination, affecting both aerial and root biomass. The saprophytic fungi *C. rigida* could have a high tolerance to potential pollutants in the sludge and a high capacity to degrade organic material and increase usable nutrient forms for the plants as seen for other saprophytes used in bioremediation [20,53,54]. These findings underscore the intricate interaction between microbial inoculation and sludge dose, suggesting that exceeding optimal doses may negatively impact plant growth, warranting careful consideration in agricultural and forestry practices. This could be explained by the presence of contaminant and other toxic compounds in the sludge [55] that can exceed the physiological limit of the plants and/or the capacity of the fungi to sequester them of make them harmless. Furthermore, comparison with the control treatment, which received no microbial application, highlights the pivotal role of MF and SF microorganisms in enhancing growth performance when utilizing residual sludge as biofertilizers in crops [56]. The significantly lower growth observed in the control group underscores the potential of microbial inoculation to maximize the utilization of nutrient resources present in sludge, thereby driving plant growth and productivity [48,57,58].

Previous research has shown synergistic effects of some saprophytic fungi on plant growth and root colonization by arbuscular mycorrhizal fungi in soil contaminated with heavy metals [42,43,59,60]. However, other studies have found neutral results in plant growth due to the combined action of arbuscular mycorrhizal fungi and *Trichoderma* species [61]. Additionally, recent research by Johnson et al. [62] demonstrated the potential for synergistic interactions between specific mycorrhizal fungi and *Trichoderma* strains in improving plant growth under stress conditions. Moreover, García-Seco et al. [63] demonstrated that the combined inoculation of arbuscular mycorrhizal fungi and *Trichoderma* spp. resulted in improved plant growth and nutrient uptake in tomato plants under saline conditions. Similarly, Singh et al. [64] showed enhanced plant tolerance to heavy metal stress through the combined application of mycorrhizal fungi and phosphate-solubilizing bacteria. Furthermore, contrasting results have been reported regarding the impact of mycorrhizal associations on plant resistance to biotic stressors. While some studies have indicated a positive effect of mycorrhizal colonization on plant defense mechanisms against pathogens [65,66,67], others have found limited, neutral, or even negative responses [68,69]. These discrepancies underscore the need for further research to elucidate the underlying mechanisms governing the interactions between mycorrhizal fungi, saprophytic microorganisms, and plant hosts under varying environmental conditions and highlight the complexity of underground interactions.

The final mycorrhization state achieved by *Quillaja saponaria* plants in the treatments involving the MF *G. roseae* and the MF + SF combination presents intriguing findings. Specifically, plants treated solely with *G. roseae* exhibited the highest degree of mycorrhization when no residual sludge was applied [42]. However, a notable decrease in mycorrhization was observed as the dose of residual sludge increased. These findings suggest a nuanced relationship between microbial inoculation, sludge dose, and mycorrhization in *Q. saponaria* plants. Studies have shown that the addition of organic fertilizer can either reduce or increase the growth of arbuscular mycorrhizal fungi [70,71,72]. The observed decrease in mycorrhization with increasing sludge doses in the MF treatment group may indicate potential interference or inhibition of mycorrhizal colonization processes when exposed to higher sludge concentrations [73]. Most studies have found that sewage sludge reduces both the pre-symbiotic and in-plant stages of the development of mycorrhizal fungus [74,75]. In contrast, the positive response to increasing sludge doses in the MF + SF combination treatment group suggests a synergistic effect between the two types of fungi, with the saprophytic fungi potentially enhancing mycorrhization [76], even under higher sludge concentrations. In addition to arbuscular mycorrhizal fungi, saprophytic fungi are another important group of microorganisms that can provide energy for other microorganisms, including arbuscular mycorrhizal fungi, by breaking down cellulosic materials into simple sugars [77]. Saprophytic fungi can also increase the metabolic activity of arbuscular mycorrhizal fungi inside the root in the presence of sewage sludge [43].

Soil enzymes such as fluorescein diacetate hydrolase (FDA), β-glucosidase, phosphatase, and dehydrogenase can be used as indicators of soil quality and soil microbial activity. This is because these enzymes are highly sensitive to alterations in soil management [78]. Here, enzymatic activity, particularly laccase and MnP, increased in plants inoculated with the SF and with up to 75% of the sludge. However, at 100% of sludge, both enzymes were not detected, which may be due to the fact that very few fungi produce laccase and MnP under excess nitrogen [79]. In ex vitro conditions, rhizospheric microorganisms favored FDA activity. However, this activity decreased when a 100% dose of the sludge was applied. For β-glucosidase, results were less clear. This enzyme was generally unaffected by fungal inoculation but increased with a 75% sludge dose in the SF+MF treatment. This suggests that some interaction between the saprophyte and the mycorrhizal fungi is occurring in the presence of the sludge, likely mediated by an increase in available resources. These results agree with previous data evidencing the positive effect of soil fungi and sludge addition on soil enzymatic activity [43,60,80].

As mentioned above, the study reveals that while moderate sludge doses (up to 75%) improve plant growth and enzymatic activity, higher doses (100%) can have detrimental effects. These findings highlight the importance of balancing nutrient inputs to avoid exceeding the physiological tolerance of plants or the functional capacity of microbial communities to detoxify contaminants [31,81]. From an ecological perspective, moderate sludge application aligns with the principles of sustainable soil management by enhancing soil fertility, promoting microbial activity, and supporting plant growth in degraded or nutrient-poor soils [26,28]. However, excessive sludge doses can introduce toxic compounds, such as heavy metals or organic pollutants, that accumulate in the soil and may pose long-term ecological risks, including reduced soil biodiversity, altered microbial communities, and impaired ecosystem functions [30,32]. These findings emphasize the need for carefully calibrated sludge application strategies to maximize benefits while minimizing environmental risks [82].

Overall, these results underscore the complex interactions between microbial inoculation, soil amendments, and plant-microbe interactions in shaping mycorrhization dynamics in plants. They also highlight the necessity of biological information on plant–fungi interactions in waste managing practices. Further research is needed to elucidate the underlying mechanisms driving these observed patterns and optimize microbial inoculation strategies for maximizing mycorrhizal symbiosis, nutrient uptake, and plant growth in agricultural and forestry systems. Additionally, investigating the long-term effects of sludge application on mycorrhization and plant health would provide valuable insights into sustainable soil management practices [83].

## 5. Conclusions

The findings of this study highlight the potential of microbial inoculation and residual sludge application in enhancing plant growth and mycorrhization dynamics in *Q. saponaria,* an endemic Chilean tree. Our results demonstrate that the application of both saprophytic and mycorrhizal fungi, either individually or in combination, positively influences plant height, diameter, and mycorrhization degree. Notably, the interaction between microbial inoculation and sludge dose revealed complex dynamics, with optimal growth observed at intermediate sludge doses and potential inhibition at higher doses.

Furthermore, comparisons with control treatments underscored the significant role of microbial inoculation in maximizing nutrient utilization and driving plant growth, particularly when utilizing residual sludge as biofertilizers in agricultural systems.

## Figures and Tables

**Figure 1 jof-11-00002-f001:**
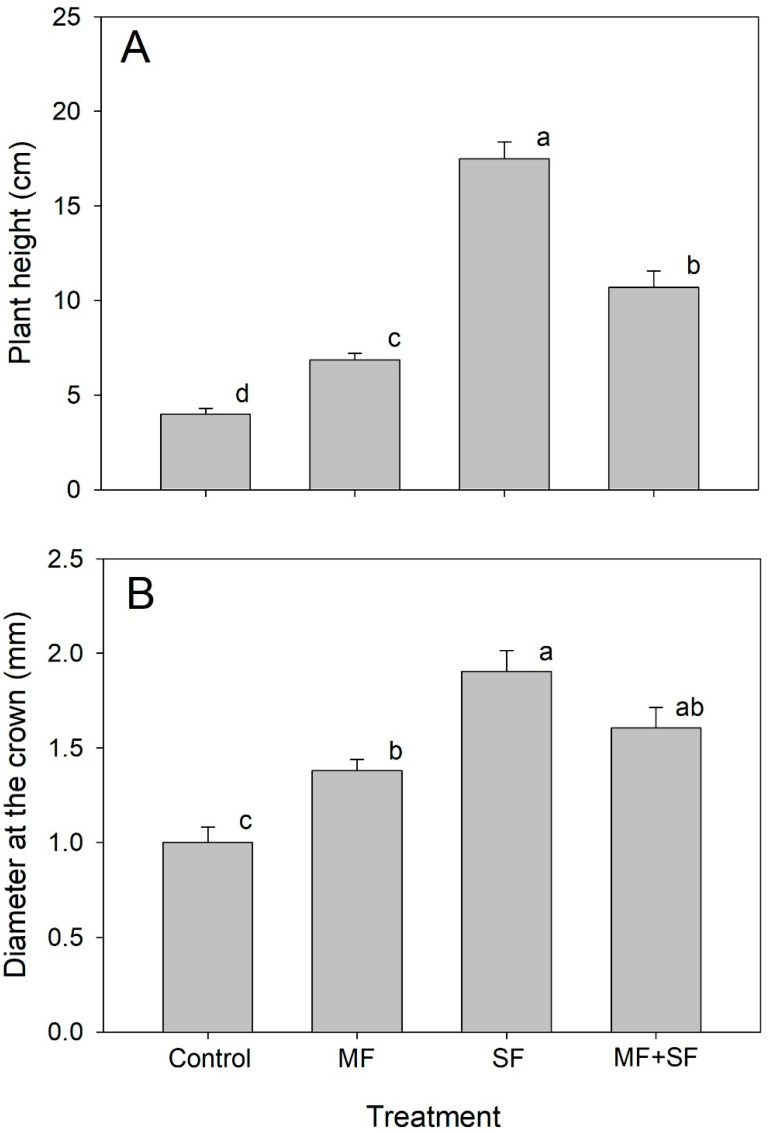
Growth performance of *Q. saponaria* plants inoculated with mycorrhizal and saprophytic fungi in sandy substrate. Height growth (**A**) and DAC (**B**) of *Q. saponaria* plants inoculated with the MF *G. roseae* and the SF *C. rigida*, and with both (MF + SF) in sandy substrate. Different letters denote significant differences (Tukey test *p* < 0.05).

**Figure 2 jof-11-00002-f002:**
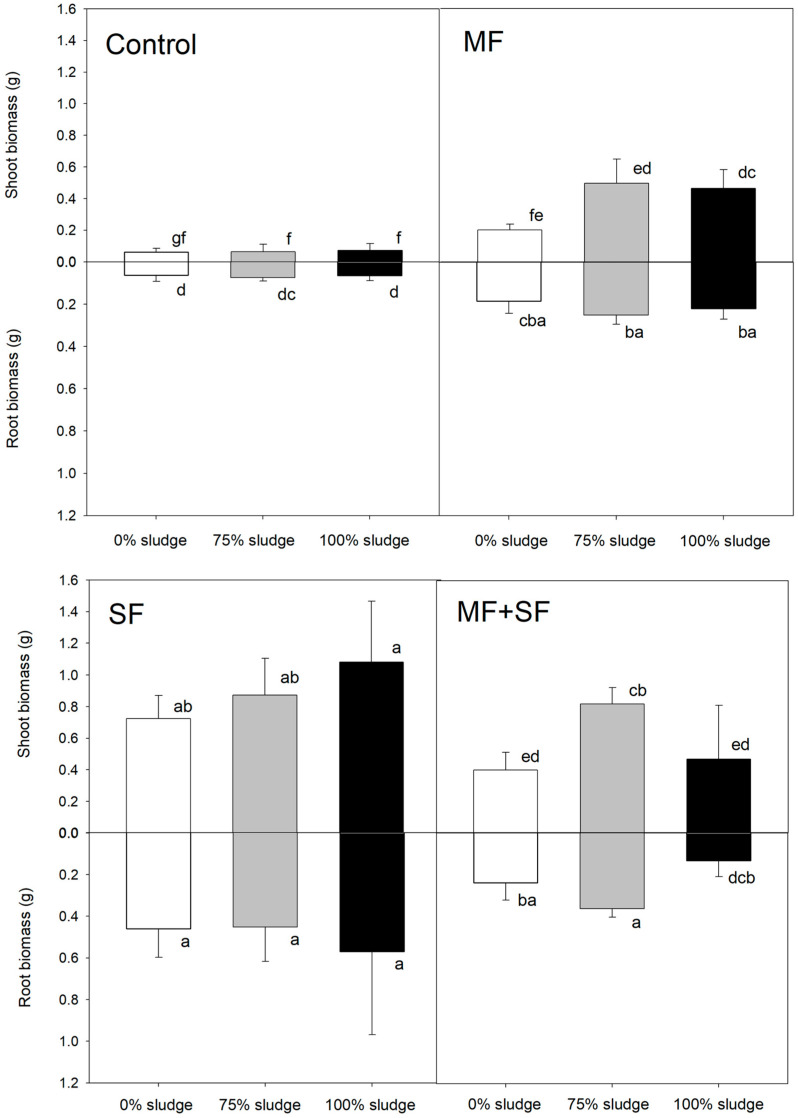
Aerial and radicle biomass of *Q. Saponaria* plants inoculated with the MF *G. roseae* and SF *C. rigida*, or their combination (MF + SF), under increasing doses of residual sludge. Different letters denote significant differences (Tukey test *p* < 0.05).

**Figure 3 jof-11-00002-f003:**
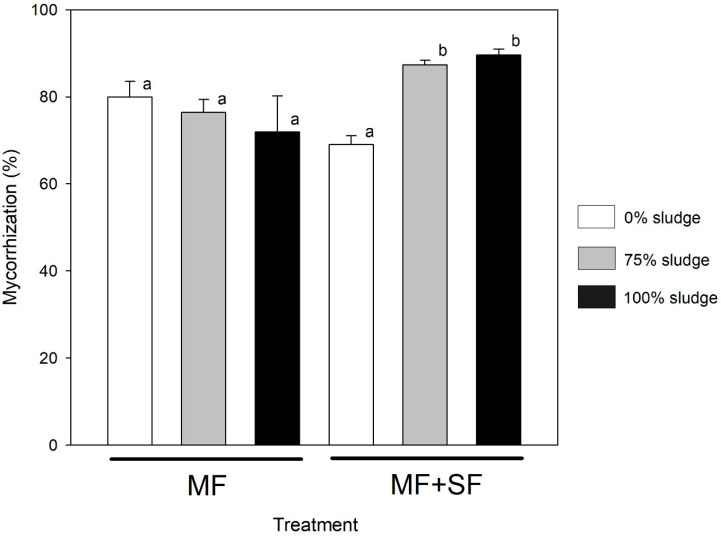
Mycorrhization % in *Q. saponaria* plants inoculated with the MF *G. roseae* or with a combination of MF + SF (*G. roseae* and *C. rigida*) after the addition of increasing concentrations of waste sludge. Different letters denote significant differences (Tukey test *p* < 0.05).

**Table 1 jof-11-00002-t001:** Chemical composition of sewage sludge utilized for plant cultivation.

Parameter	Units	Organic Sewage Sludge
pH (H_2_O)	-	6.99
Organic matter	%	52.1
Kjeldahl nitrogen	(g kg^−1^)	38.7
Total P	(mg kg^−1^)	1279
Available K	(mg kg^−1^)	3079
Iron	(mg kg^−1^)	283.7
Aluminum extractable	(mg kg^−1^)	88
Cadmium	(mg kg^−1^)	2.83
Nickel	(mg kg^−1^)	28.5
Lead	(mg kg^−1^)	60.9
Copper	(mg kg^−1^)	737.1
Zinc	(mg kg^−1^)	821.9

**Table 2 jof-11-00002-t002:** Soil chemical characteristics before and after application with sewage sludge.

Parameter	Sandy Soil	Soil + 75% Sludge	Soil + 100% Sludge
pH (H_2_O)	6.74	6.62	6.63
Organic matter (%)	0.03	0.18	0.20
Kjeldahl nitrogen	6.0	10.4	11.2
Olsen-P (mg kg^−1^)	0.3	3.7	4.1
Available-K (mg kg^−1^)	83.9	62.9	70.9
Exchangeable Al (mg kg^−1^)	0.02	0.01	0.02
Exchangeable Mg (mg kg^−1^)	0.16	0.14	0.17
CEC (cmol kg^−1^)	1.67	2.29	2.42
Fe (mg kg^−1^)	13.4	18.2	20.2
Mn (mg kg^−1^)	0.9	1.9	2.2
Zn (mg kg^−1^)	0.1	0.3	0.3
Cu (mg kg^−1^)	0.4	0.5	0.5

**Table 3 jof-11-00002-t003:** Enzymes activities (*Saprobe* fungi in vitro) U nk^−1^.

Treatments	Enzymes	0%	75%	100%
*Coriolopsis rigida*	Laccase	2.912	3.65	ND
	Mn-Peroxidase	0.88	0.95	ND

ND: Not detected.

**Table 4 jof-11-00002-t004:** Enzymes activities FDA (sewage sludge incubated with saprobe fungi) μg fluorescein g^−1^. Lower-case letters indicate a significant difference between inoculation treatments and upper-case letters indicate a significant difference between sewage sludge doses (Tukey test, *p* < 0.05).

Treatments	AM Inoculation	0%	75%	100%
None	None	9.2 a A	12.4 a B	12.3 a B
	*Gigaspora roseae*	28.4 c C	31.2 c D	20.3 c A
*Coriolopsis rigida*	None	14.1 b B	12.7 a A B	14.5 a b B
	*Gigaspora roseae*	48.3 d C	24.0 b A B	20.5 c A

**Table 5 jof-11-00002-t005:** Enzymes activities β-glucosidase (sewage sludge incubated with saprobe fungi) μmol PNP g^−1^ h^−1^. Lower-case letters indicate a significant difference between inoculation treatments and upper-case letters indicate a significant difference between sewage sludge doses *p* < 0.05. (Tukey test, *p* < 0.05).

Treatments	AM Inoculation	0%	75%	100%
None	None	2.13 a A	3.08 a B	2.72 a A B
	*Gigaspora roseae*	2.46 b A	3.27 a b B C	2.96 a b B
*Coriolopsis rigida*	None	2.17 a A	3.56 c B	3.49 c B
	*Gigaspora roseae*	1.99 a A	4.18 d C	3.16 b B

## Data Availability

The data that supports the findings of this study is available from the corresponding author upon reasonable request.

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
