# Peer review of "Gigaspora roseae and Coriolopsis rigida Fungi Improve Performance of Quillaja saponaria Plants Grown in Sandy Substrate with Added Sewage Sludge"

_jof, 2024, doi:10.3390/jof11010002_

Round 1
Reviewer 1 Report
In general, the manuscript is interesting and highlights the importance of the use of microorganisms such as arbuscular mycorrhizal fungi and saprophytic fungi as enhancing agents in the growth and development of plants of interest.
On the other hand, I highlight some comments or suggestions that could help to further improve your manuscript.
1. In the introduction section, you highlight the importance and advantages of using mycorrhizal fungi for the sequestration of heavy metals and, thus reducing heavy metal toxicity, however, you do not describe what positive impacts the use of saprophytic fungi has for the reduction of contaminants, so I recommend including a short paragraph that gives context as to why saprophytic fungi were used and what benefits can be obtained from them as you do describing mycorrhizal fungi (line 57 to line 69).
2. Add to the list of references the citation Eivazi and Tabatabai (1990) (line 103).
3. Add to the citation Adam and Duncan (2021) the numerical citation as it does in the citation Saparrat and Guillen (21) (line 106).
4. Change the units cc to cm3 on line 113 and line 116.
5. In line 107 make use of superscripts in the units of measurement (M-1 cm-1).
6. On line 192 change the symbol “%” to percentaje (mycorrhizal colonization % change to mycorrhizal colonization percentaje).
7. From line 211 to line 215 make use of super indexes in the units of measurement (kg-1).
1. Add to the list of references the citation Eivazi and Tabatabai (1990) (line 103).
2. Add to the citation Adam and Duncan (2021) the numerical citation as it does in the citation Saparrat and Guillen (21) (line 106).
3. Change the units cc to cm3 on line 113 and line 116.
4. In line 107 make use of superscripts in the units of measurement (M-1 cm-1).
5. On line 192 change the symbol “%” to percentaje (mycorrhizal colonization % change to mycorrhizal colonization percentaje).
6. From line 211 to line 215 make use of super indexes in the units of measurement (kg-1).
Author Response
In general, the manuscript is interesting and highlights the importance of the use of microorganisms such as arbuscular mycorrhizal fungi and saprophytic fungi as enhancing agents in the growth and development of plants of interest.
On the other hand, I highlight some comments or suggestions that could help to further improve your manuscript.
We greatly thank reviewer 1 for all comments and suggestions that will greatly improve our ms
- In the introduction section, you highlight the importance and advantages of using mycorrhizal fungi for the sequestration of heavy metals and, thus reducing heavy metal toxicity, however, you do not describe what positive impacts the use of saprophytic fungi has for the reduction of contaminants, so I recommend including a short paragraph that gives context as to why saprophytic fungi were used and what benefits can be obtained from them as you do describing mycorrhizal fungi (line 57 to line 69).
Information on saprophytic fungi and their potential in bioremediation was included at the end of the second paragraph of the introduction (Please see Lines 63 to 76).
- Add to the list of references the citation Eivazi and Tabatabai (1990) (line 103).
Added as suggested
- Add to the citation Adam and Duncan (2021) the numerical citation as it does in the citation Saparrat and Guillen (21) (line 106).
Corrected in the text. Adam and Duncan is now cited correctly.
- Change the units cc to cm3 on line 113 and line 116.
Done as suggested
- In line 107 make use of superscripts in the units of measurement (M-1cm-1).
Done as suggested
- On line 192 change the symbol “%” to percentaje (mycorrhizal colonization % change to mycorrhizal colonization percentaje).
Corrected as suggested
- From line 211 to line 215 make use of super indexes in the units of measurement (kg-1).
Corrected as suggested
Reviewer 2 Report
The manuscript explores the use of fungi and residual sewage sludge to enhance the growth of Quillaja saponaria. This topic is relevant for sustainable agriculture and waste management, providing innovative approaches to reuse organic waste. While the manuscript is a significant contribution to the field, it requires improvements in methods clarity, figure presentation, and deeper interpretation of certain results to fully meet publication standards.
1. The context for choosing Quillaja saponaria as a model plant is clear, but it would benefit from a more explicit justification of why its saponin content is ecologically or industrially relevant to this study.
2. Were spore concentrations or fungal colony-forming units quantified before application? If not, how was inoculum consistency ensured across replicates?
3. The factorial design is appropriate; however, the manuscript does not detail how potential interactions between sludge doses and fungi inoculation were hypothesized.
4. While enzyme activities are reported, the study does not explain the absence of Mn-peroxidase and lacase activity at 100% sludge.
5. The discussion thoroughly compares findings with previous research but does not delve deeply into the ecological implications of the observed sludge dose thresholds.
1. Line 21-23: The statement about the high content of saponins in Quillaja saponaria is informative but lacks context.
2. Line 152-156: Significant differences in plant height are mentioned, but the magnitude of the differences is not provided in the text.
3. Fig. 3: The interaction between sludge dose and mycorrhization is reported but not fully interpreted.
4. Line 237-239: The conclusion about C. rigida producing the highest biomass under 100% sludge dose is compelling but lacks mechanistic support.
5. Line 274-278: The reduced mycorrhization at higher sludge doses is mentioned but not fully linked to potential causes.
6. Fig 1: The figure shows plant height but does not indicate the percentage increase compared to the control.
7. Table 5: The enzymatic activity data for β-glucosidase lacks clarity on why it increases with specific treatments.
Author Response
The manuscript explores the use of fungi and residual sewage sludge to enhance the growth of Quillaja saponaria. This topic is relevant for sustainable agriculture and waste management, providing innovative approaches to reuse organic waste. While the manuscript is a significant contribution to the field, it requires improvements in methods clarity, figure presentation, and deeper interpretation of certain results to fully meet publication standards.
1. The context for choosing Quillaja saponaria as a model plant is clear, but it would benefit from a more explicit justification of why its saponin content is ecologically or industrially relevant to this study.
This was included in the last paragraph of the introduction (Please see Lines 91 to 119). We thanks for the comment since we believe it would clarify the selection of the model species.
2. Were spore concentrations or fungal colony-forming units quantified before application? If not, how was inoculum consistency ensured across replicates?
We rephrased the inoculum section in the methods to make it clearer (Please see Lines L159 to 168). Additionally, we cite several reference to justify the inoculum method and we used previously described approaches for the mycorrhizal and the saprophytic fungi. Lastly, the inoculation method was consistently applied among all sludge treatments, so minor error will be equally distributed.
3. The factorial design is appropriate; however, the manuscript does not detail how potential interactions between sludge doses and fungi inoculation were hypothesized.
This is briefly mention at the end of the introduction (Please see Lines 114 to 117) and discussed later (page 13, Line 296 to 319).
4. While enzyme activities are reported, the study does not explain the absence of Mn-peroxidase and laccase activity at 100% sludge.
This is addressed in Lines 342 to 344.
5. The discussion thoroughly compares findings with previous research but does not delve deeply into the ecological implications of the observed sludge dose thresholds.
This was added to the discussion section (Please see Lines 354 to 364).
Specific comments
1. Line 21-23: The statement about the high content of saponins in Quillaja saponaria is informative but lacks context.
We believe that this is resolved with the addition the paragraph in the introduction regarding saponins (see response to comment 1).
2. Line 152-156: Significant differences in plant height are mentioned, but the magnitude of the differences is not provided in the text.
The % change was added to the results as suggested.
3. Fig. 3: The interaction between sludge dose and mycorrhization is reported but not fully interpreted.
We believe that this is addressed Please see Lines 378-396
4. Line 237-239: The conclusion about C. rigida producing the highest biomass under 100% sludge dose is compelling but lacks mechanistic support.
A possible mechanistic explanation was added in the first paragraph of the discussion (Please see Lines 321 to 340).
5. Line 274-278: The reduced mycorrhization at higher sludge doses is mentioned but not fully linked to potential causes.
This is now clearer with the addition of a paragraph in the discussion (Please see Lines 357 to 369).
6. Fig 1: The figure shows plant height but does not indicate the percentage increase compared to the control.
The % change was added to the results as suggested in the first paragraph of the results. We believe, however, that this data can be deducted from Figure 1.
7. Table 5: The enzymatic activity data for β-glucosidase lacks clarity on why it increases with specific treatments.
We changed the redaction in the text to make it clearer and we added statistical data to Table 5 (Please see Lines 265 to 26 and Table 5).
Round 2
Reviewer 2 Report
My suggested changes have been incorporated
none